# Clinical characteristics and factors associated with long COVID among post-acute COVID-19 clinic patients in Zambia, August 2020 to January 2023: A cross-sectional and longitudinal study design

**Warren Malambo**[1]*, **Duncan Chanda**[2,3], **Lily Besa**[3], **Daniella Engamba**[2,3], **Linos Mwiinga**[1], **Mundia Mwitumwa**[2], **Peter Matibula**[2], **Neil Naik**[2], **Suilanji Sivile**[2,3], **Simon Agolory**[1], **Andrew Auld**[1], **Lloyd Mulenga**[3], **Jonas Z. Hines**[1☯], **Sombo Fwoloshi**[2,3☯]

1 Division of Global HIV and TB, U.S. Centers for Disease Control and Prevention, Global Health Center, Lusaka, Zambia, 2 Department of Internal Medicine, University Teaching Hospital Adult and Emergency Hospital, Lusaka, Zambia, 3 Ministry of Health, Lusaka, Zambia

☯ These authors contributed equally to this work.

* ykh1@cdc.gov

**Data Availability Statement:** Data are not currently publicly available but may be obtained by a third

## Abstract

### Introduction

A number of seroprevalence studies in Zambia document the extent of spread of acute SARS-CoV-2 infection, yet knowledge gaps still exist on symptoms and conditions that continue or develop after acute COVID-19 (long COVID). This is an important gap given the estimated prevalence of long COVID in other African countries. We assessed factors associated with long COVID at the initial visit to a post-acute COVID-19 (PAC-19) clinic and longitudinally among a cohort of patients with ≥2 review visits.

### Methods

We implemented a cross-sectional and longitudinal analysis of PAC-19 clinic patients from Aug-2020 to Jan-2023. The study outcome was long COVID; defined as the presence of new, relapsing, or persistent COVID-19 symptoms that interfere with the ability to function at home or work. Explanatory variables were demographic and clinical characteristics of patients which included sex, age group, presence of new onset medical conditions, presence of pre-existing comorbidities, vaccination status and acute COVID-19 episode details. We fitted logistic and mixed effects regression models to assess for associated factors and considered statistical significance at p<0.05.

### Results

Out of a total 1,359 PAC-19 clinic patients in the cross-sectional analysis, 548 (40.3%) patients with ≥2 PAC-19 clinic visits were in the longitudinal analysis. Patients' median age was 53 (interquartile range [IQR]: 41–63) years, 919 (67.6%) were hospitalized for acute

party. The data are de-identified participant data, available with permission from the Government of the Republic of Zambia – Ministry of Health (MoH). A request to access the data can be made to the Permanent Secretary - Technical Services (ps@moh.gov.zm), Zambia Ministry of Health, Ndeke House Haile Selassie Ave, P.O Box 30205, Lusaka 10101, Zambia, who together with technical leads will review the request and avail the data. Protocols and statistical analysis information is available as per above.

**Funding:** This study has been supported in part by the President's Emergency Plan for AIDS Relief (PEPFAR) through the Centers for Disease Control and Prevention (CDC). Grant number/CoAg ID number: GH002234. The study sponsor or funder had no role, in the study design; in the collection, analysis, and interpretation of data; in the writing of the report; and in the decision to submit the article for publication. In addition, there is independence of researchers from funders and all authors, external and internal, had full access to all of the data (including statistical reports and tables) in the study and can take responsibility for the integrity of the data and the accuracy of the data analysis. There was no additional external funding received for this study.

**Competing interests:** The authors declare that they have no known competing financial interests or personal relationships that could have appeared to influence the work reported in this paper.

COVID-19, and of whom 686 (74.6%) had severe acute COVID-19. Overall, 377 (27.7%) PAC-19 clinic patients had long COVID. Patients with hospital length of stay ≥15 days (adjusted odds ratio [aOR]: 5.37; 95% confidence interval [95% CI]: 2.99–10.0), severe acute COVID-19 (aOR: 3.22; 95% CI: 1.68–6.73), and comorbidities (aOR:1.50; 95% CI: 1.02–2.21) had significantly higher chance of long COVID. Longitudinally, long COVID prevalence significantly (p<0.001) declined from 75.4% at the initial PAC-19 visit to 26.0% by the final visit. The median follow-up time was 7 (IQR: 4–12) weeks.

## Conclusion

Factors associated with long COVID in Zambia were consistent both cross-sectionally at the initial visit to PAC-19 clinics and longitudinally across subsequent review visits. This highlights the importance of ongoing monitoring and tailored interventions for patients with comorbidities and severe COVID-19 to mitigate the long-term impacts of COVID-19.

## Introduction

Since the identification of the first SARS-CoV-2 infections in December 2019, the Coronavirus disease 2019 (COVID-19) has become a major public health problem. Its morbidity and mortality dramatically increased across countries and regions leading to its classification as a pandemic [1]. As of February 2024, the World Health Organization (WHO) had reported over 774 million cumulative COVID-19 cases and over 7 million cumulative COVID-19 deaths globally [2]. In the Africa region alone, over 9.5 million cumulative COVID-19 cases and over 175 thousand COVID-19 deaths were reported for the same time period.

People infected with SARS-CoV-2 commonly develop symptoms within 4–5 days following exposure although others may be asymptomatic [3]. Acute COVID-19 (within 4 weeks from the initial SARS-CoV-2 infection) presents with as a multi-system cluster of symptoms, i.e., general, cardiovascular, pulmonary, gastrointestinal, neurologic, musculoskeletal, and ear, nose, and throat. Recovery from COVID-19 for most patients occurs within 7–10 days after symptoms onset but could take weeks to months in some patients [4–7]. Previous epidemics such the 2003 severe acute respiratory syndrome (SARS) and the 2012 Saudi Arabia Middle East respiratory syndrome coronavirus (MERS-CoV) had patients who presented with persistent symptoms post-acute infection [8–11]. Their symptoms included fatigue, decreased quality of life, shortness of breath and behavior health problems that impacted on their health-related quality of life.

Symptoms and conditions that continue or develop after acute COVID-19 infection are variably named (post-acute COVID-19, post-acute sequelae of SARS-CoV-2 infection, post-COVID conditions, long-haul COVID, long-term effects of COVID and chronic COVID) but are also referred to as long COVID [6, 12]. Commonly reported long COVID symptoms include fatigue, fever, cough, dizziness, brain fog, and myalgia although symptoms are varied and potentially have overlapping etiologies [13–15]. Long COVID is thus a syndrome characterized by persistent, new, relapsing or delayed SAR-CoV-2 symptoms ≥12 weeks beyond onset of the acute episode [12, 16–18], and can have a protracted path to recovery that impacts on quality of life, earnings and health care costs [19–21].

As much as 77 million people around the world could be estimated to have long COVID, based on a conservative estimated prevalence of 10% of reported number of infected people

[2, 22]; the actual number may likely be much higher given undocumented cases due to limited testing capacity. Among acute-COVID-19 outpatient cases, the prevalence of long COVID is estimated at 10–30%; 50–70% of hospitalized cases and 10–12% of vaccinated cases [22–27]. For example, a study of the burden, causation, and particularities of long COVID in 7 African countries estimated a pooled long COVID prevalence of 41% [28]. Long COVID is associated with increasing age, comorbidities, hospitalization for acute COVID-19, severe COVID-19, and vaccination status [29–31]. Vaccination against SARS-CoV-2 is, however, associated with reduced likelihood of severe COVID-19 and long COVID.

There's growing evidence on long COVID from countries in Africa [28, 32–34]. Disease severity and admission to the intensive care unit during acute COVID-19, for example, were found to be associated with long COVID in Nigeria and South Africa [33, 35]. Scoping studies on long COVID in Africa found comorbidities and age >40 years to be associated factors [33, 36]. In Zambia, 17% of persons with acute COVID-19 in July 2020 were found to be experiencing symptoms ~2 months later [37]. Zambia's reported COVID-19 cases are, however, likely an underestimate given limited testing capacity, asymptomatic infections, and mild clinically ill cases who may not have come to the attention of the health system [38–40].

Whereas a number of studies in Zambia document the extent of spread and associated factors of COVID-19, Knowledge gaps still exist regarding long-term outcomes of SARS-CoV-2 infection [35, 37, 38, 40–45]. For example, although a previous cohort study investigated persistent post-acute COVID-19 symptoms, comparisons were not made across SARS-CoV-2 variants nor was the effect of vaccination assessed [37]. Similarly, another study on the time-to-recovery from COVID-19 only considered associated factors among acute hospitalized patients and not those related to post-acute sequelae of SARS-CoV-2 infection [45]. This is an important gap given the estimated prevalence and burden of long COVID in other African countries. In this study, we thus considered acute COVID-19 discharged patients presenting for follow up care in specialized post-acute COVID-19 (PAC-19) clinics to describe long COVID in Zambia. We assessed factors associated with long COVID at initial visit to a PAC-19 clinic and longitudinally among a cohort of patients with ≥2 PAC-19 clinic visits.

## Methods

### Study setting

Beginning August 2020, the Zambia Ministry of Health set up 13 specialized PAC-19 clinics to care for people following SARS-CoV-2 infection. PAC-19 clinics were set up initially at the two national referral hospitals (University Teaching and Levy Mwanawasa Teaching hospitals) in Lusaka city, the capital of Zambia, and subsequently at 11 other major teaching and general hospitals across all 10 provinces in Zambia.

### Study participants

Per national PAC-19 clinical guidelines, patients diagnosed with SARS-CoV-2 infection of all ages presented in PAC-19 clinics for follow-up care after discharge from hospital admission or outpatient episodic [46]. At the initial PAC-19 clinic visit, patients' demographics, medical history, comorbidities, and current symptoms were recorded on a standardized paper form. Physical examinations and laboratory investigations were conducted based on patient's medical history as required. Patients' ability to perform activities, as part of examination of their functional and mental health status, were also evaluated.

Further clinical appointments of up to 5 review visits were at the clinicians' discretion. Symptoms were assessed by clinicians at each PAC-19 clinic visit on review of systems: general, cardiovascular, pulmonary, urinary, neurologic, musculoskeletal, ear, nose, and throat (ENT),

mental health concerns, depression, anxiety, and dermatologic. Pre-existing comorbidities (hypertension, diabetes, cardiovascular disease, cancer, chronic lung disease, kidney disease, or liver disease, immunosuppression, obesity, HIV and TB) were documented. Patients requiring further specialist care were referred to various units including physiotherapy, cardiology, endocrinology, nephrology, psychiatry, and pulmonology. Anonymized data from patients standardized clinical paper forms were routinely abstracted into an electronic database (RED-Cap v11.0.3); which was accessed on 5th January 2023 and analyzed for this study.

## Study design

We implemented a cross-sectional and longitudinal study design to assess for factors associated with long COVID at initial visit to a PAC-19 clinic and longitudinally across review visits. First, we performed a cross-sectional analysis of all PAC-19 clinic patients with data entered in the REDCap electronic databases from August 5, 2020, through to January 26, 2023. We then did a longitudinal sub-analysis of patients with ≥2 PAC-19 clinic review visits.

## Study variables

The study outcome was long COVID; a binary compound variable defined as the presence of persistent COVID-19 symptoms that interfere with the ability to function at home or work [47]. Functional status since acute COVID-19 diagnosis was assessed as changes in ability to self-care (including washing oneself or getting dressing), day-to-day work/school, taking care of household tasks, standing for >30 minutes, walking long distances (>1 km), concentrating for ten minutes, mood and sleep pattern. Assessment for persistent symptoms were based on clinicians' review of systems at each PAC-19 clinic visit. New onset medical conditions were also assessed and included hypertension, diabetes mellitus, cardiovascular diseases, and stroke. Due to limited diagnostic capacity in PAC-19 clinics, tissue injury of multiple organs that has the potential to lead to long-term organ dysfunction were not assessed as one of the delineations for long COVID [47].

Explanatory variables included demographic and clinical characteristics of PAC-19 clinic patients which included sex, age group, presence of new onset disease, presence of pre-existing comorbidities, vaccination status and acute COVID-19 episode details (hospitalization status, hospital length of stay, and severe acute COVID-19 status). Severe acute COVID-19 was a binary variable defined as COVID-19 episode that required supplemental oxygen therapy, intensive care unit (ICU) stay or treatment with steroids or remdesivir. Vaccination status was also binary categorized based on vaccination records, when available, or patients' self-reported status. SARS CoV-2 variants (Wild type, Beta, Delta and Omicron) classification was based on the dominant variant at the time of SARS-CoV-2 diagnosis as captured by the Zambia's genomic surveillance system and submitted to the Global Initiative on Sharing All Influenza Data (GISAID) rather than sequenced specimens from patients [48].

At longitudinal sub-analysis, the data structure included repeated observations of long COVID symptoms. Demographic and clinical characteristics of patients included time-invariant covariates sex, age group (patients repeatedly observed for <1 year), presence of new onset medical conditions, presence of pre-existing comorbidities, and acute COVID-19 episode details. Time-variant covariates included COVID-19 vaccination status and referral to specialist services.

## Study size

For the study sample size, we assumed prevalence of long COVID at 30% and a precision corresponding to the effect size of 5%. The prevalence or Cochrane formula (described elsewhere)

was used to calculate the minimum sample size of 323 [49]. Study participants were included in the study if they were COVID-19 patients presenting in PAC-19 clinics for follow-up care after discharge from hospital or outpatient episodic care and had their information abstracted to the clinics' REDCap electronic database. A full enumeration of all PAC-19 clinic attendees was however considered for higher accuracy and precision. For the longitudinal sub-analysis, patients were excluded from the analysis if they attended a PAC-19 clinic only once.

## Statistical analysis

At cross-sectional analysis, descriptive statistics were reported as frequencies with proportions for categorical variables and medians with the interquartile range (IQR) for non-normally distributed numeric variables assessed using the Shapiro-Wilk test. The Pearson Chi-square and Kruskal Wallis tests for proportions and Wilcoxon rank sum test for medians were used to determine the association of explanatory variables to the outcome. An unadjusted and adjusted logistic regression model was fitted to assess for factors associated with having long COVID at initial visit to a PAC-19 clinic.

The main analysis was the cross-sectional study of acute COVID-19 hospitalized patients (inpatients) to maintain key variables (severe COVID-19 and hospital length of stay). At multivariable analysis, we adjusted for age, presence of new onset medical conditions, presence of pre-existing comorbidities, and COVID-19 episode details (i.e., hospital length of stay, and presence of severe COVID-19). Covariate inclusion criteria at multivariable analysis were based on theoretical relevance to the study and $p < 0.2$. Statistical significance was, however, considered at $p < 0.05$. We also conducted an additional cross-sectional analysis that included both inpatients and outpatients during acute infection to assess for association of hospitalization status during acute COVID-19 with long COVID.

For the longitudinal sub-analysis, we fitted a mixed effects model to account for the heterogeneity in long COVID i.e., the differences between patients and within patients across repeated observations or PAC-19 clinic visits. An unadjusted and adjusted mixed effects model was separately fitted for acute COVID-19 inpatients only and additionally for both inpatients and outpatients. The patient was the random effects term for the model. Since there was dependence in the data, we also reported the conditional intraclass correlation (ICC) which quantified the variance in long COVID heterogeneity that was attributable to between patients and within patients across PAC-19 visits. Analysis was done in R software version 4.3.2 (R Foundation for Statistical Computing).

## Bias

To control for bias, only covariates with less than 10% missingness were included at multivariable analyses. We assessed for potential bias in estimates by checking for the proportion of listwise deletion when covariates of interest with >10% missingness were included at multivariable analysis. Estimates with listwise deletion greater than 50% was considered as biased. Covariates such as vaccination status and referral to specialist services with high proportion of missing and listwise deletion were thus not included at multivariable analysis.

## Ethical considerations

This study was preconceived during the development of clinical and operational guidelines for the management of PAC-19 patients in Zambia. The study obtained ethical clearance (waiver for informed consent) from the University of Zambia Biomedical Research Ethics Committee (Ref No. 2711–2022), approval from the Zambia National Health Research Authority (Ref No: NHRA0002/26/05/2022) and was determined to be non-research according to the U.S Centers

for Disease Control and Prevention (CDC) policy and applicable federal law (See e.g., 45 C.F. R. part 46.102(l)(2), 21 C.F.R. part 56; 42 U.S.C. §241(d); 5 U.S.C. 552a; 44 U.S.C.3501 et seq.). CDC investigators did not interact with patients or have access to personally identifiable information but participated in protocol development and analysis of anonymized data.

## Results

Out of a total 1,359 PAC-19 clinic patients in the cross-sectional analysis (Fig 1), 548 (40.3%) patients with ≥2 PAC-19 clinic visits were included in the longitudinal sub-analysis. Overall, the patients' median age was 53 (IQR: 41–63) years, and 693 (52.9%) were female. Most patients (58.0%) were from Lusaka, where the first two PAC-19 clinics were initially set up (Table 1). Nearly half of the patients (46.9%) were diagnosed with COVID-19 when delta was the dominant variant (Fig 2). The median time since COVID-19 diagnosis when patients presented for care in PAC-19 clinics was 4 (IQR:2–6) weeks. Of all patients, most (n = 919, 84.5%) were hospitalized during acute COVID-19.

At baseline, 654 (48.1%) patients overall, reported having had pre-existing comorbidity. Among these patients, hypertension (n = 465, 71.1%), diabetes (n = 172, 26.3%), HIV (n = 165, 25.2%) and cardiovascular disease (n = 73, 11.2%) were commonly (frequency of ≥10%) reported (S1 Table). One hundred nineteen (8.8%) patients had new onset medical conditions

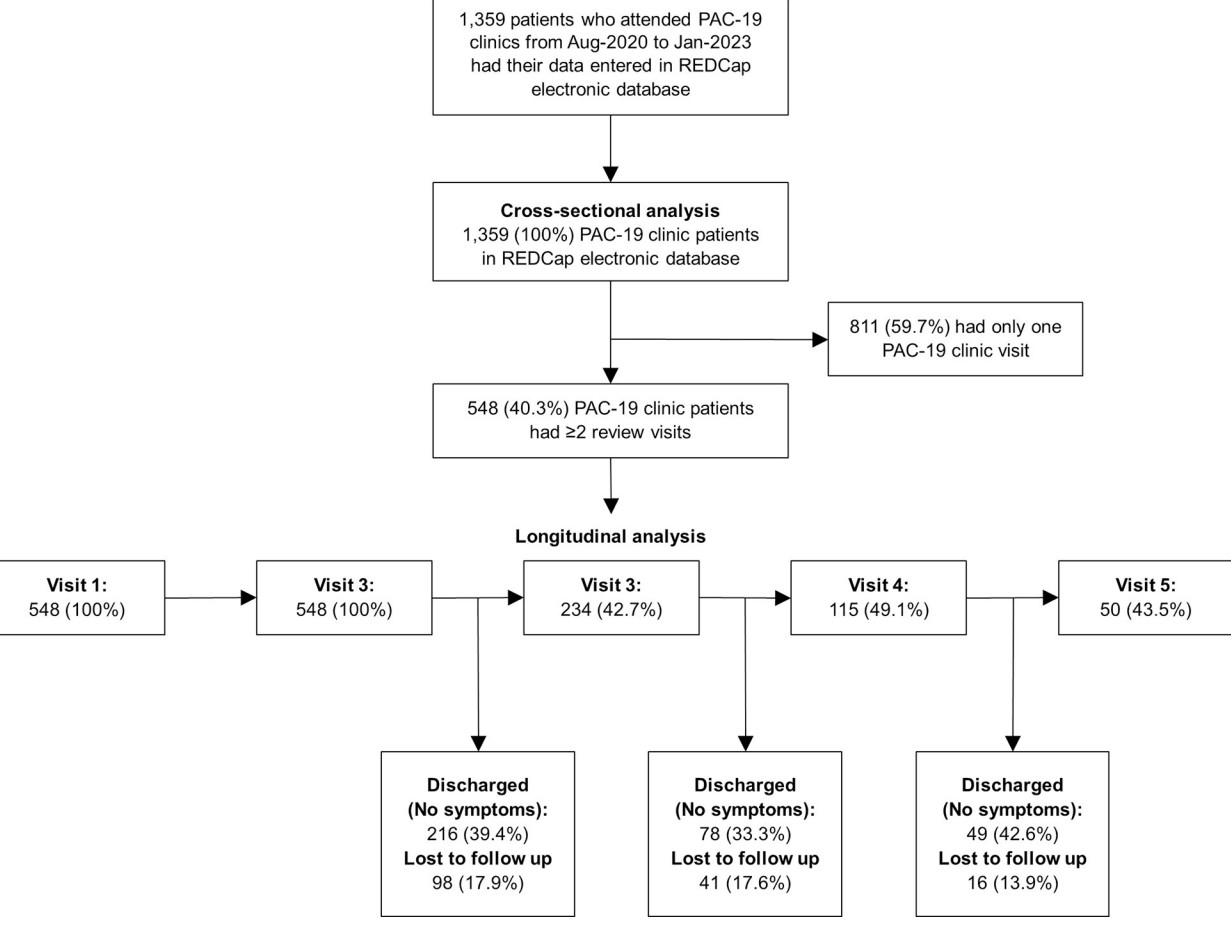

**Fig 1. Analysis flow diagram of PAC-19 clinic patients in Zambia, August 2020 –January 2023.**

**Table 1. Cross-sectional demographic and clinical characteristics of PAC-19 clinics patients in Zambia, Aug. 2020–Jan. 2023 (N = 1,359).**

| Characteristic | Presence of long COVID symptoms | | | p-value [a] |
|---|---|---|---|---|
| | Yes (377); n (%) | No (982); n (%) | Overall; n (%) | |
| **Sex** | | | | 0.994 |
| Female | 197 (53.0) | 496 (52.9) | 693 (52.9) | |
| Male | 175 (47.0) | 441 (47.1) | 616 (47.1) | |
| (Missing) | 5 | 45 | 50 | |
| **Age (years)** | | | | **0.016** |
| Median (IQR) | 53 (43, 66) | 52 (40, 63) | 53 (41, 63) | |
| (Missing) | 0 | 45 | 45 | |
| **Age group (years)** | | | | **0.039** |
| ≤29 | 22 (5.8) | 98 (10.5) | 120 (9.1) | |
| 30–39 | 47 (12.5) | 132 (14.1) | 179 (13.6) | |
| 40–49 | 84 (22.3) | 188 (20.1) | 272 (20.7) | |
| 50–59 | 82 (21.8) | 219 (23.4) | 301 (22.9) | |
| 60+ | 142 (37.7) | 300 (32.0) | 442 (33.6) | |
| (Missing) | 0 | 45 | 45 | |
| **PAC-19 Clinic location** | | | | 0.152 |
| Lusaka city | 158 (54.5) | 433 (59.4) | 591 (58.0) | |
| Other districts | 132 (45.5) | 296 (40.6) | 428 (42.0) | |
| (Missing) | 87 | 253 | 340 | |
| **Dominant SARS-CoV-2 variant at diagnosis [b]** | | | | **<0.001** |
| Wild type | 17 (4.6) | 55 (5.8) | 72 (5.4) | |
| Beta | 94 (25.3) | 186 (19.5) | 280 (21.1) | |
| Delta | 190 (51.1) | 433 (45.3) | 623 (46.9) | |
| Omicron | 71 (19.1) | 282 (29.5) | 353 (26.6) | |
| (Missing) | 5 | 26 | 31 | |
| **Time since COVID-19 diagnosis (weeks) [c]** | | | | **<0.001** |
| Median (IQR) | 6 (5, 10) | 3.0 (2, 5) | 4.0 (2, 6) | |
| (Missing) | 118 | 251 | 369 | |
| **Presence of pre-existing comorbidities [d]** | | | | **<0.001** |
| No | 155 (41.1) | 550 (56.0) | 705 (51.9) | |
| Yes | 222 (58.9) | 432 (44.0) | 654 (48.1) | |
| **Presence of new onset medical conditions [e]** | | | | **0.005** |
| No | 331 (87.8) | 909 (92.6) | 1,240 (91.2) | |
| Yes | 46 (12.2) | 73 (7.4) | 119 (8.8) | |
| **Vaccination status** | | | | 0.422 |
| Not vaccinated | 250 (78.1) | 543 (75.8) | 793 (76.5) | |
| Vaccinated | 70 (21.9) | 173 (24.2) | 243 (23.5) | |
| (Missing) | 57 | 266 | 323 | |
| **Hospitalization status during acute COVID-19** | | | | 0.428 |
| Outpatient | 45 (14.1) | 123 (16.0) | 168 (15.5) | |
| Inpatient | 274 (85.9) | 645 (84.0) | 919 (84.5) | |
| (Missing) | 58 | 214 | 272 | |
| **Hospitalization length of stay (days) [f]** | | | | **<0.001** |
| 1–3 | 19 (9.4) | 139 (26.8) | 158 (21.9) | |
| 4–7 | 38 (18.8) | 165 (31.8) | 203 (28.2) | |
| 8–14 | 48 (23.8) | 125 (24.1) | 173 (24.0) | |
| ≥15 | 97 (48.0) | 90 (17.3) | 187 (25.9) | |

*(Continued)*

**Table 1.** (Continued)

| Characteristic | Presence of long COVID symptoms | | | p-value [a] |
|---|---|---|---|---|
| | Yes (377); n (%) | No (982); n (%) | Overall; n (%) | |
| *(Missing)* | 72 | 126 | 198 | |
| **Severe COVID [f, g]** | | | | **0.001** |
| No | 50 (18.2) | 183 (28.4) | 233 (25.4) | |
| Yes | 224 (81.8) | 462 (71.6) | 686 (74.6) | |
| **Functional status since COVID-19** | | | | **<0.001** |
| Better | 144 (55.2) | 427 (78.8) | 571 (71.1) | |
| Same/worse [h] | 117 (44.8) | 115 (21.2) | 232 (28.9) | |
| (Missing) | 116 | 440 | 556 | |
| **Referral to specialist services [i]** | | | | **0.001** |
| No | 221 (89.1) | 614 (95.0) | 835 (93.4) | |
| Yes | 27 (10.9) | 32 (5.0) | 59 (6.6) | |
| (Missing) | 129 | 336 | 465 | |

[a] Pearson Chi-square and Kruskal Wallis tests were used for proportions and Wilcoxon rank sum test for medians. Bolded p-values are significant at p<0.05

[b] SARS CoV-2 variants classification based on GISAID (2023) rather than sequenced specimens from study patients

[c] Computed as a difference in weeks between date of diagnosis and date of presentation at PAC-19 clinic

[d] Pre-existing comorbidity: hypertension, diabetes, cardiovascular disease, cancer, immunosuppression, chronic lung, kidney, and liver diseases, obesity, HIV and TB.

[e] Comorbidities (hypertension, diabetes, and HIV) diagnosed at the time of SARS CoV-2 infection.

[f] Denominator is 919 (i.e., patients hospitalized for acute COVID-19)

[g] Acute COVID-19 episode that required supplemental oxygen therapy, intensive care unit stay or treatment with steroids or remdesivir

[h] Same or worse limitations in daily activities since acute COVID-19

[i] Specialist services included physiotherapy, cardiology, endocrinology, nephrology, psychiatry, and pulmonology

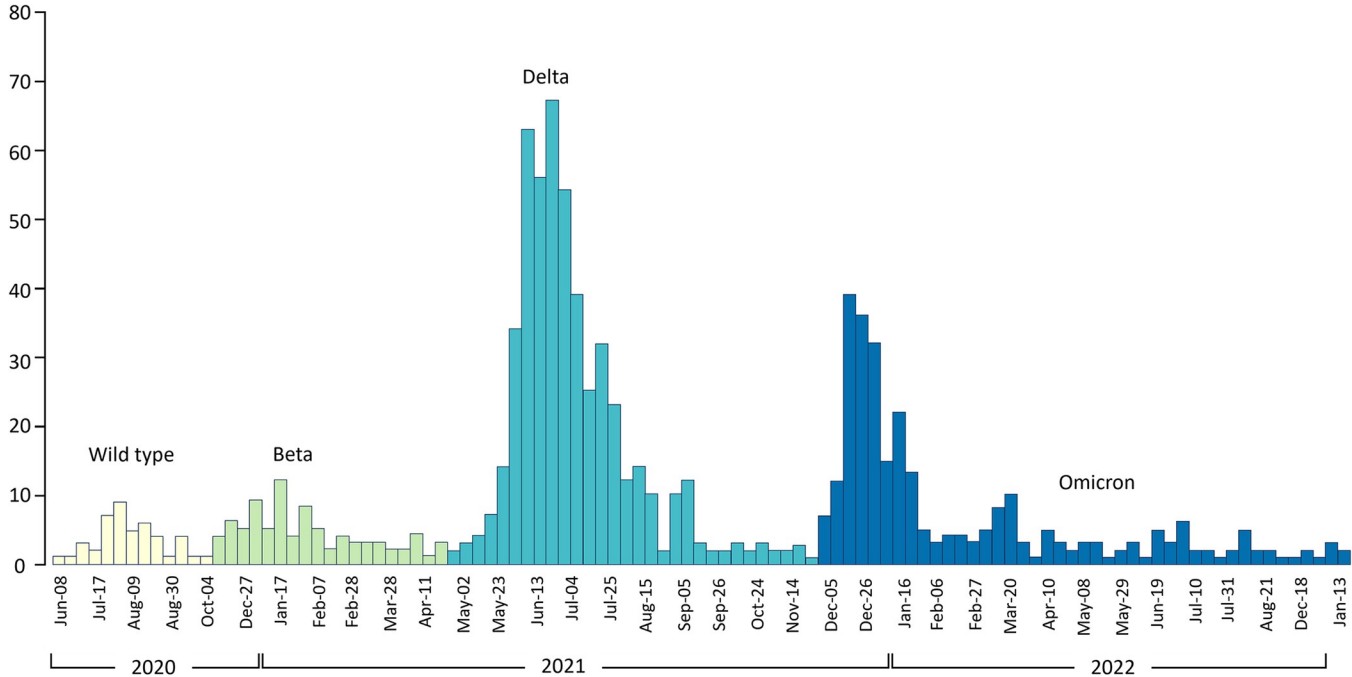

**Fig 2. PAC-19 clinic patients' attendance by date of diagnosis, Zambia: August 2020 –January 2023.**

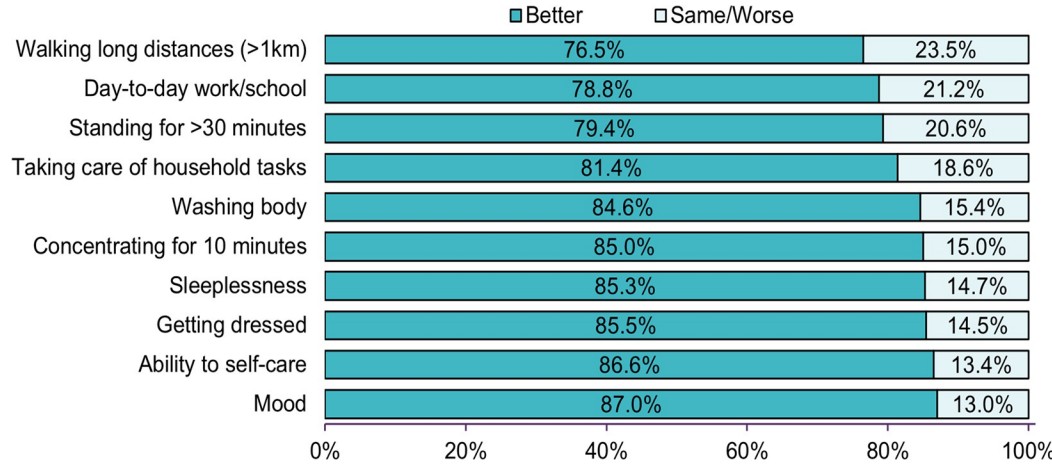

**Fig 3. PAC-19 clinic patients' self-reported functional status over a 7-day period preceding visit to PAC-19 clinic in comparison to since acute COVID-19.**

at the time of SARS-CoV-2 infection. Of these patients, new onset hypertension and diabetes were detected in 68 (53.8%) and 64 (53.8%) patients, respectively.

Among the hospitalized acute COVID-19 patients, the median length of stay was 7 days (IQR:4–15 days). Of these inpatients, 686 (74.6%) were categorized as having had severe acute COVID-19. The medium time from the date of COVID-19 diagnosis to when patients presented for care in a PAC-19 clinic was 4 weeks (IQR: 2–6 weeks). At initial visit to a PAC-19 clinic, 274 (29.8%) acute COVID-19 hospitalized PAC-19 clinic patients had long COVID. Among these long COVID patients, commonly reported symptoms included cough (38.7%), fatigue (38.5%), shortness of breath (26.5%), chest pain (20.9%), headache (14.8%), muscles aches/pain (14.1%), palpitations (12.5%), joint aches/pain (11.9%), and forgetfulness or brain fog (8.0%). Among the outpatients, 103 (23.4%) had long COVID, bringing the overall number of long COVID patients to 377 (27.7%).

Of all patients, 232 (17.1%) had worse or same (limitations in daily activities) functional status since acute COVID-19 (Fig 3). The functional status patients commonly reported difficulty in undertaking was walking long distances greater than 1km (23.5%), day-to-day work/school (cognitive) activities (21.2%), standing for ≥30 minutes (20.6%), taking care of household tasks (18.6%), and self-care (activities like bathing and dressing)– 15.4%. Fifty-nine (25.4%) of PAC-19 clinic patients functional limitation were referred to specialist care, including cardiology (35.6%), endocrinology (33.9%), psychiatry (25.4%), pulmonology (18.6%), physiotherapy (11.0%), and nephrology (1.7%).

Overall, 243 (23.5%) patients presented to PAC-19 clinics having been vaccinated prior to being diagnosed with SARS-CoV-2 (Table 1 and S3 Table), representing 28.5% of patients seen at PAC-19 clinics from April 2021 when COVID-19 vaccines first became available in Zambia. Of these vaccinated patients, 102 (42%) received Johnson and Johnson's Janssen (Ad26.COV2.S), 97 (39.9%) received AstraZeneca (AZD1222), 41 (16.9%) did not know the vaccine type they received, and 3 (1.2%) received Pfizer-BioNTech (BNT162b2). Fifty-seven (23.4%) had received a full series of vaccines >14 days prior to their date of SARS-CoV-2 diagnosis, 31 (12.7%) were partially vaccinated (i.e., 2nd dose not yet received), 10 (4.1%) were diagnosed with SARS-CoV-2 within 14 days of being vaccinations while 148 (60.9%) had missing vaccination dates to be classified.

**Table 2. Cross-sectional analysis of factors associated with long COVID among acute COVID-19 hospitalized PAC-19 clinic patients in Zambia, August 2020 to January 2023 (N = 919).**

| Variable | Unadjusted OR (95% CI) | p-value [a] | Adjusted OR (95% CI) | p-value [a] |
|---|---|---|---|---|
| **Sex** | | | | |
| Female | *Referent* | - | Referent | - |
| Male | 0.98 (0.74, 1.31) | 0.912 | 0.92 (0.65–1.32) | 0.667 |
| **Age group (years)** | | | | |
| ≤29 | *Referent* | - | *Referent* | - |
| 30–39 | 1.73 (0.88, 3.51) | 0.116 | 0.87 (0.38, 2.01) | 0.741 |
| 40–49 | 2.01 (1.09, 3.87) | **0.029** | 0.82 (0.39, 1.76) | 0.596 |
| 50–59 | 1.69 (0.93, 3.24) | 0.096 | 0.70 (0.33, 1.51) | 0.350 |
| 60+ | 1.86 (1.05, 3.47) | **0.041** | 0.61 (0.30, 1.28) | 0.181 |
| **Presence of pre-existing comorbidities [b]** | | | | |
| No | *Referent* | | *Referent* | - |
| Yes | 1.50 (1.12, 2.01) | **0.006** | 1.50 (1.02, 2.21) | **0.041** |
| **Presence of new onset medical conditions [c]** | | | | |
| No | *Referent* | - | *Referent* | - |
| Yes | 1.48 (0.94, 2.30) | 0.084 | 1.34 (0.78, 2.28) | 0.284 |
| **Hospital length of stay (days)** | | | | |
| 1–3 | *Referent* | - | *Referent* | - |
| 4–7 | 1.68 (0.94, 3.11) | 0.086 | 1.26 (0.68, 2.39) | 0.478 |
| 8–14 | 2.81 (1.59, 5.14) | **<0.001** | 1.98 (1.08, 3.74) | **0.031** |
| ≥15 | 7.88 (4.60, 14.1) | **<0.001** | 5.37 (2.99, 10.0) | **<0.001** |
| **Severe COVID-19 [d]** | | | | |
| No | *Referent* | | *Referent* | - |
| Yes | 1.77 (1.26, 2.54) | **0.001** | 3.22 (1.68, 6.73) | **<0.001** |
| **Vaccination status [e]** | | | | |
| Not vaccinated | *Referent* | | - | - |
| Vaccinated | 0.94 (0.64, 1.36) | 0.737 | - | - |
| **Referral to specialist services [e]** | | | | |
| No | *Referent* | | - | - |
| Yes | 1.88 (1.00, 3.48) | **0.045** | - | - |

Abbreviation: OR—odds ratio; CI–confidence interval

[a] Bolded p-values are significant at p<0.05

[b] Pre-existing comorbidity: hypertension, diabetes, cardiovascular disease, cancer, immunosuppression, chronic lung, kidney, and liver diseases, obesity, HIV and TB.

[c] Comorbidities (hypertension, diabetes, and HIV) diagnosed at the time of SARS CoV-2 infection.

[d] Acute COVID-19 episode that required supplemental oxygen therapy, intensive care unit stay or treatment with steroids/ remdesivir

[e] Vaccination status and referral to specialist services were not adjusted for at multivariable analysis due to >10% missingness which resulted in 50.5% listwise deletion.

At cross-sectional analysis (Table 2), PAC-19 clinic patients with pre-existing comorbidities (adjusted odds ratio [aOR]: 1.50; 95% confidence interval [CI]: 1.02–2.21), and with hospital length of stay of 8–14 and ≥15 days were associated with significantly higher chance of long COVID (aOR: 1.98; CI: 95% 1.08–3.74, and aOR: 5.37; 95% CI: 2.99–10.0, respectively). Patients who had severe COVID-19 had 3.23-fold (95% CI: 1.68–6.75) higher chance of presenting with long COVID. Independently, vaccinated patients had non-significant reduced odds ratio of long COVID (OR: 0.94; 95% CI: 0.64–1.36) while those who were referred to specialist services had 1.88-fold (95% CI: 1.00–3.48; p-value = 0.045) increased chance of long COVID.

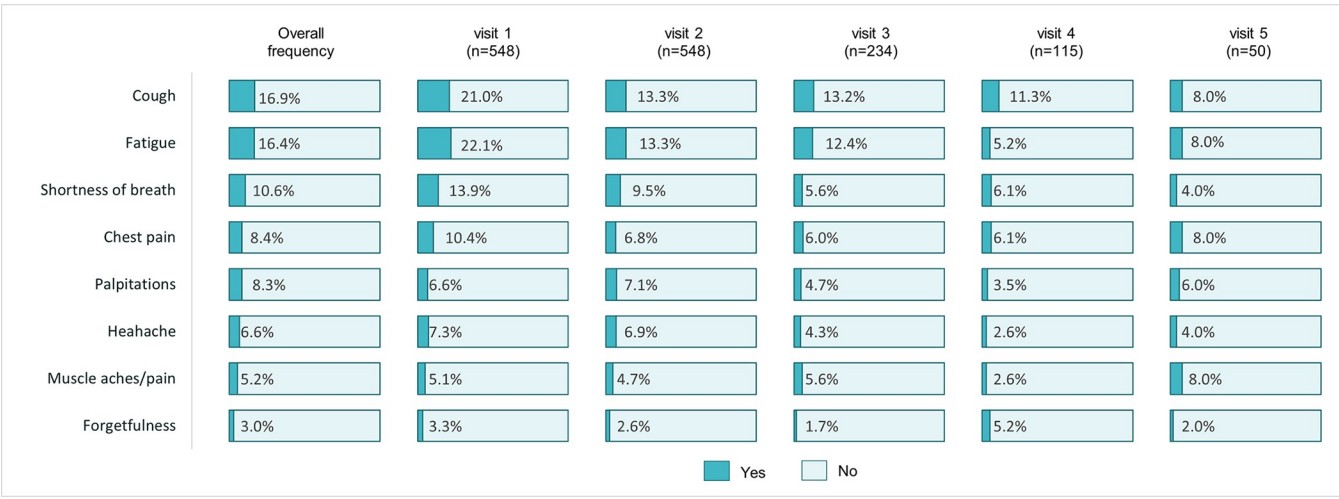

**Fig 4. Longitudinal frequency of most common long COVID symptoms among PAC-19 clinic patients in Zambia, Aug. 2020 -Jan. 2023.**

In the additional cross-sectional analysis that included both inpatients and outpatients during acute COVID-19 (S2 Table), inpatients were associated with non-significant increased chance of long COVID (aOR: 1.05; 95% CI: 0.72–1.55). Patients with the presence of comorbidities were, however, associated with significantly higher chance of long COVID (aOR: 1.55; 95% CI: 1.16–2.08) while the presence of new onset medical conditions had a non-significant increase odds ratio of long COVID. Patients aged 40–49 years were significantly associated with 1.79-fold increased chance of long COVID (95% CI: 1.03–3.22). Independently, patients referred to specialist services were 2.34 times (95% CI: 1.37–4.00) more likely to present with long COVID. Males compared to females and having been vaccinated against COVID-19 were associated with non-significant reduced chance of long COVID.

In the longitudinal analysis (S1 Fig), the overall median time patients presented at a PAC-19 clinic was 7 (IQR: 4–12) weeks. Longitudinally (Fig 4), the most frequently (≥5% at any PAC-19 clinic visit) reported long COVID symptoms were cough (16.9%), fatigue (16.4%), shortness of breath (10.6%), chest pain (8.4%), headache (8.3%), palpitations (6.6%) and muscle aches/pains (5.2%), and forgetfulness (3.0%). The prevalence of long COVID significantly declined (p<0.001) from 75.4% at the initial PAC-19 clinic visit to 53.5% at the second visit, 59.8% at the third visit, 49.6% at the fourth visit, and 26.0% at the final visit (S3 Table). Overall, the proportion of vaccinated patients attending PAC-19 clinics increased from 20% at the initial visit to 58% at the fifth visit; an increase that was statistically significant (p<0.001). Similarly, comparatively significantly (p = 0.002) larger proportion of patients with comorbidities presented in subsequent PAC-19 clinic visits than at the initial PAC-19 clinic visit. The proportion of patients with new onset medical conditions longitudinally declined; this was, however, non-significant (p = 0.953).

We found similar associations with long COVID in the longitudinal analysis (Table 3) as with the cross-sectional analysis (Table 2). Across subsequent clinical visits, the odds ratio of presenting with long COVID declined overall. Patients with hospital length of stay of ≥15 days and those with severe COVID-19 had significantly increased chance of presenting with long COVID (aOR: 4.30; 95% CI: 1.54–12.0, and aOR: 1.89; 95% CI: 1.02–3.49, respectively). Longitudinally, vaccinated patients had non-significant reduced odds ratio of long COVID (aOR: 0.78; 95% CI: 0.45–1.35). Similarly, sex, age, presence of comorbidities and new onset medical conditions were associated with non-significant odds ratio of long COVID.

**Table 3. Longitudinal mixed effects analysis of factors associated with long COVID among acute COVID-19 hospitalized PAC-19 clinic patients in Zambia, August 2020 to January 2023 (N = 540).**

| Variable | Adjusted model [a] | | Fully adjusted model [b] | |
|---|---|---|---|---|
| | aOR (95% CI) | p-value [c] | aOR (95% CI) | p-value [c] |
| **PAC-19 clinic visit** | | | | |
| 1 | Referent | - | Referent | - |
| 2 | 0.18 (0.12, 0.27) | **<0.001** | 0.18 (0.11, 0.31) | **<0.001** |
| 3 | 0.09 (0.04, 0.18) | **<0.001** | 0.09 (0.04, 0.21) | **<0.001** |
| 4 | 0.02 (0.01, 0.06) | **<0.001** | 0.02 (0.01, 0.06) | **<0.001** |
| 5 | 0.00 (0.00, 0.01) | **<0.001** | 0.00 (0.00, 0.02) | **<0.001** |
| **Sex** | | | | |
| Female | Referent | - | Referent | - |
| Male | 0.81 (0.52, 1.27) | 0.367 | 0.75 (0.42, 1.36) | 0.348 |
| **Age group (years)** | | | | |
| ≤29 | Referent | - | Referent | - |
| 30–39 | 2.25 (0.64, 7.94) | 0.208 | 1.07 (0.19, 6.09) | 0.937 |
| 40–49 | 1.60 (0.51, 5.02) | 0.422 | 0.68 (0.14, 3.26) | 0.634 |
| 50–59 | 1.56 (0.51, 4.81) | 0.438 | 0.79 (0.17, 3.75) | 0.765 |
| 60+ | 2.56 (0.85, 7.67) | 0.094 | 1.27 (0.27, 5.84) | 0.762 |
| **Presence of new onset medical conditions** | | | | |
| No | Referent | - | Referent | - |
| Yes | 1.55 (0.76, 3.14) | 0.227 | 1.41 (0.58, 3.43) | 0.449 |
| **Presence of comorbidities** | | | | |
| No | Referent | - | Referent | - |
| Yes | 1.86 (1.18, 2.92) | **0.007** | 1.66 (0.88, 3.14) | 0.117 |
| **Hospital length of stay (days)** | | | | |
| 1–3 | Referent | - | Referent | - |
| 4–7 | 1.95 (0.72, 5.25) | 0.187 | 1.73 (0.65, 4.61) | 0.271 |
| 8–14 | 2.18 (0.80, 5.90) | 0.127 | 1.74 (0.64, 4.71) | 0.277 |
| ≥15 | 5.92 (2.14, 16.42) | **0.001** | 4.30 (1.54, 12.0) | **0.005** |
| **Severe COVID-19 [d]** | | | | |
| No | Referent | - | Referent | - |
| Yes | 1.44 (0.90, 2.31) | 0.124 | 1.89 (1.02, 3.49) | **0.042** |
| **Vaccination status** | | | | |
| Not vaccinated | Referent | - | Referent | - |
| Vaccinated | 0.79 (0.50, 1.26) | 0.326 | 0.78 (0.45, 1.35) | 0.380 |
| **Referral to specialist Services [e]** | | | | |
| No | Referent | - | - | - |
| Yes | 1.01 (0.53, 1.95) | 0.973 | - | - |
| **Random Effects** | | | | |
| Between PAC-19 clinic visit variance | | | 3.29 | |
| Within PAC-19 clinic visit variance | | | 4.22 | |
| Conditional intraclass correlation | | | 0.56 | |

[a] Models adjusted for fixed effects term PAC-19 clinic visit occasion.

[b] Fully adjusted model included measurement occasion fixed effects term (PAC-19 clinic visit occasions) and patient level fixed effects terms: age (<29, 30–39, 40–49, 50–59, or +60 years), presence of comorbidities (yes or no), presence of new onset comorbidity (yes or no), acute COVID-19 hospitalization length of stay (1–3, 4–7, 8–14, and ≥15 days), severe COVID-19 during acute COVID-19 (yes or no), and COVID-19 vaccination status (not vaccinated or vaccinated). The random-effects term for longitudinal clustering of repeated measure was the patient.

[c] Bolded p-values are significant at p<0.05

[d] Acute COVID-19 episode that required supplemental oxygen therapy, intensive care unit stay or treatment with steroids/remdesivir

[e] Referral to specialist services not adjusted for at multivariable analysis due to 32.6% missingness which resulted in 57.9% listwise deletion.

On the random effects part of the longitudinal model, we estimated a conditional ICC of 0.562. This suggested that 56.2% of the longitudinal variance in long COVID was attributable to between patient differences. This indicates a moderate level of consistency or reliability in measurements within each patient across their PAC-19 clinic review visits.

In the longitudinal sub-analysis that included both those that were inpatients and outpatients during acute COVID-19 (S4 Table), similar associations were observed as to the cross section sub-analysis (S2 Table). The odds ratio of presenting with long COVID significantly (p<0.001) declined, overall, across subsequent PAC-19 clinic visits (S4 Table). The presence of comorbidities were longitudinally associated with increased odds ratio of long COVID by a factor of 1.67 (95% CI: 1.02–2.75). Hospitalized patients during acute COVID-19, vaccinated and male patients were associated with non-significant reduced likelihood of long COVID. Similarly, order age groups and presence of new onset medical conditions were associated with non-significant, however increased, odds of long COVID.

## Discussion

Factors associated with long COVID in Zambia were similar cross-sectionally at initial visit to PAC-19 clinics and longitudinally across review visits. Associated factors at cross-sectional analysis included presence of comorbidities, hospital length of stay and severe COVID-19. Longitudinally, PAC-19 clinic review visit, hospital length of stay and severe COVID-19 were associated factors. Hospitalization for acute COVID-19 was not significantly associated with long COVID. Patients with limitation in their functional status that required referral to specialist services were independently associated with long COVID at cross-sectional analysis. Longitudinally, vaccinated patients had indistinguishable chance of presenting with long COVID.

At cross-sectional analysis, pre-existing comorbidities were associated with long COVID as previously reported [50]. This is expected given that pre-existing comorbidities are primarily related to the underlying health status of individuals and can influence the body's response to infection. Comparatively however, new onset medical conditions had a non-significant association to long COVID. The new onset medical conditions could have been as a result of multiorgan effects or autoimmune conditions [3]. Furthermore, fewer PAC-19 clinic patients, overall, had new onset medical conditions which could explain the non-significant association. Sex and age group were similarly not associated with long COVID which seem to suggest that factors associated with long COVID in Zambia were largely clinical as opposed to demographic.

Patients with longer hospital stays may have received intensive medical interventions and additional treatments. This could have potentially affected the immune response and the body's ability to recover from the acute COVID-19, which in turn may have increased the chances of long COVID symptoms. Furthermore, patients with pre-existing comorbidities may have been more susceptible to severe COVID-19 and may thus have been at higher risk of developing long COVID. This is consistent with what is reported by other studies on patients with longer hospitalization stay for acute COVID-19 or pre-existing comorbidities and their association to long COVID [51–53].

A severe form of acute COVID-19 is known to increase the risk of an intense immune response that could lead to widespread inflammation [54]. The inflammatory response can persist post-acute infection and could have led to the immune system being dysregulated, leading to prolonged persistent symptoms and complications of COVID-19 in the patients. This could have led to observed significant association of severe COVID-19 long COVID as has been reported in other studies [33, 35, 51, 55].

In both the cross-sectional and longitudinal analysis, inpatients compared to outpatient care during COVID-19 episode were found to be associated with indistinguishable odds ratio of long COVID. One reason for this might be that those who were outpatient and attending PAC-19 clinics were a special subset of outpatients with COVID-19 (i.e., they were seeking out or were referred for follow-up care during acute COVID-19e). Furthermore, inpatients (the majority of whom had severe COVID-19) might have been more likely to be referred to PAC-19 clinic whereas those who were outpatient were likely self-referral to PAC-19 clinic meaning only those who really needed it (i.e., those with ongoing symptoms) were in this study.

PAC-19 clinic patients with worse/same functional status since COVID-10 and were referred to specialist services may have had residual organ damage from acute COVID-19; although this could not be ascertained due to limited diagnostic capacity. As previously reported, this may have been closely related to the observed limitations in daily activities that impact on patients' functional status since acute COVID-19 [9]. Although this association was independently significant, we could not adjust for it at multivariable analysis due to missingness that resulted in listwise deletion of more than half the study sample. Furthermore, no feedback systems were in place to inform PAC-19 clinics of the outcome of specialist care for the referred patients.

COVID-19 vaccines have been shown to be effective at preventing symptomatic SARS-CoV-2 infections and long COVID [56, 57]. In this study, however, vaccinated patients failed to show significant protective association between COVID vaccination and long COVID. A possible explanation for this finding might be the heterogeneity in vaccination status during the study since Zambia only began offering COVID-19 vaccination in April 2021 (i.e., right before the delta wave and ~8 months after patients began presenting for care in PAC-19 clinics) [58]. For example, among the patients diagnosed with SARS-CoV-2 during the wild type and beta dominant variants periods, prior to publicly available COVID-19 vaccination in Zambia, only one patient ever reported being vaccinated. Furthermore, a consider number of patients had not yet received a full series of vaccines prior to SAR-CoV-2 infection as was evidenced by the number that got vaccinated during follow up.

With regards to recovery time from long COVID, most patients with long COVID had symptom resolution by the second month of follow up. Overall, we found that commonly reported long COVID symptoms in Zambia were similar to findings in other studies [33, 35, 37, 51]. However, symptoms like forgetfulness (commonly reported among long COVID patients) and change in sleep were less commonly reported in our study compared to previous reports [17, 20].

The findings in this study are subject to several limitations. Firstly, only patients attending PAC-19 clinics were included in the study, so other patients with long COVID who sought care elsewhere (or didn't seek care) were not included. This potentially represents some form of limitation within the study recruitment since other long COVID patients could have by-passed the PAC-19 clinic and presented directly to clinicians or specialist service at outpatient departments. Furthermore, PAC-19 clinics took time to scale-up throughout Zambia and some persons might still have faced difficulty accessing them. Overall, fewer than 1% of the over 344,000 confirmed COVID-19 cases in Zambia as of January 2023 presented for care in PAC-19 clinics.

Secondly, COVID-19 illness history (i.e., date of SARS-CoV-2 testing, hospitalization status, vaccination history, acute COVID-19 episode details, etc.) were self-reported by patients to clinicians which cannot rule out self-serving bias. Thirdly, the study utilized routinely collected clinical information from PAC-19 clinics which had substantial missingness for some key variables such as vaccination status and referral to specialist services. Thus, these variables were not adjusted for at multivariable analyses so as to maintain sample size. And lastly,

patients were not retested for SAR-CoV-2 infection at subsequent PAC-19 review visits and so the possibility of reinfection could not be ruled out.

## Conclusion

Our study found that factors associated with long COVID in Zambia were consistent both cross-sectionally at the initial visit to PAC-19 clinics and longitudinally across subsequent review visits. Cross-sectional analysis revealed that the presence of comorbidities, longer hospital stays, and severe COVID-19 were significantly associated with long COVID. Longitudinal analysis confirmed that PAC-19 clinic review visits, extended hospital stays, and severe COVID-19 were also significant factors. Interestingly, hospitalization for acute COVID-19 did not show a significant association with long COVID. Patients with functional status limitations requiring specialist referrals were independently associated with long COVID at the initial visit. Additionally, longitudinal analysis indicated that vaccinated patients had an indistinguishable chance of presenting with long COVID compared to unvaccinated patients. These findings highlight the importance of ongoing monitoring and tailored interventions for patients with comorbidities and severe COVID-19 to mitigate the long-term impacts of the COVID-19.

## Supporting information

**S1 Fig. Histogram of longitudinal PAC-19 clinic attendance (all visits) by time since COVID-19 diagnosis, August 2020 January 2023.**
(TIF)

**S1 Table. Frequencies of pre-existing and new onset comorbidities of PAC-19 clinic patients in Zambia, Aug. 2020–Jan. 2023.**
(DOCX)

**S2 Table. Factors associated with long COVID at initial visit to PAC-19 clinic among acute COVID-19 inpatients and outpatients in Zambia, August 2020 –January 2023 (N = 1,359).**
(DOCX)

**S3 Table. Longitudinal demographic and clinical characteristics of PAC-19 clinic patients in Zambia, Aug. 2020–Jan. 2023.**
(DOCX)

**S4 Table. Longitudinal risk factors and association for long COVID among acute COVID-19 hospitalized and non-hospitalized patients in Zambia, August 2020 –January 2023 (N = 548).**
(DOCX)

## Acknowledgments

**Disclaimer:** The findings and conclusions in this report are those of the authors and do not necessarily represent the official position of the US Centers for Disease Control and Prevention (CDC) or the study funders.

## Author Contributions

**Conceptualization:** Warren Malambo, Duncan Chanda, Lily Besa, Daniella Engamba, Linos Mwiinga, Mundia Mwitumwa, Peter Matibula, Neil Naik, Suilanji Sivile, Simon Agolory, Lloyd Mulenga, Jonas Z. Hines, Sombo Fwoloshi.

**Data curation:** Warren Malambo, Duncan Chanda, Lily Besa, Daniella Engamba, Linos Mwiinga, Mundia Mwitumwa, Peter Matibula, Neil Naik, Suilanji Sivile, Simon Agolory, Lloyd Mulenga, Jonas Z. Hines, Sombo Fwoloshi.

**Formal analysis:** Warren Malambo, Duncan Chanda, Lily Besa, Daniella Engamba, Linos Mwiinga, Mundia Mwitumwa, Simon Agolory, Andrew Auld, Lloyd Mulenga, Jonas Z. Hines, Sombo Fwoloshi.

**Funding acquisition:** Duncan Chanda, Linos Mwiinga, Peter Matibula, Simon Agolory, Lloyd Mulenga, Jonas Z. Hines, Sombo Fwoloshi.

**Investigation:** Duncan Chanda, Lily Besa, Daniella Engamba, Mundia Mwitumwa, Peter Matibula, Neil Naik, Suilanji Sivile, Sombo Fwoloshi.

**Methodology:** Warren Malambo, Duncan Chanda, Lily Besa, Daniella Engamba, Mundia Mwitumwa, Peter Matibula, Neil Naik, Jonas Z. Hines, Sombo Fwoloshi.

**Project administration:** Duncan Chanda, Lily Besa, Daniella Engamba, Mundia Mwitumwa, Peter Matibula, Neil Naik, Suilanji Sivile, Lloyd Mulenga, Sombo Fwoloshi.

**Resources:** Duncan Chanda, Lily Besa, Mundia Mwitumwa, Peter Matibula, Neil Naik, Sombo Fwoloshi.

**Software:** Warren Malambo, Duncan Chanda, Lily Besa, Sombo Fwoloshi.

**Supervision:** Warren Malambo, Duncan Chanda, Lily Besa, Linos Mwiinga, Mundia Mwitumwa, Peter Matibula, Neil Naik, Suilanji Sivile, Simon Agolory, Andrew Auld, Lloyd Mulenga, Jonas Z. Hines, Sombo Fwoloshi.

**Validation:** Warren Malambo, Duncan Chanda, Lily Besa, Daniella Engamba, Linos Mwiinga, Simon Agolory, Andrew Auld, Lloyd Mulenga, Jonas Z. Hines, Sombo Fwoloshi.

**Visualization:** Warren Malambo, Duncan Chanda, Linos Mwiinga, Andrew Auld, Lloyd Mulenga, Jonas Z. Hines, Sombo Fwoloshi.

**Writing – original draft:** Warren Malambo, Duncan Chanda, Lily Besa, Daniella Engamba, Linos Mwiinga, Mundia Mwitumwa, Peter Matibula, Neil Naik, Suilanji Sivile, Simon Agolory, Andrew Auld, Lloyd Mulenga, Jonas Z. Hines, Sombo Fwoloshi.

**Writing – review & editing:** Warren Malambo, Duncan Chanda, Lily Besa, Daniella Engamba, Linos Mwiinga, Mundia Mwitumwa, Peter Matibula, Neil Naik, Suilanji Sivile, Simon Agolory, Andrew Auld, Lloyd Mulenga, Jonas Z. Hines, Sombo Fwoloshi.

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
