## [Editor Report · Decision Letter 0]

17 Jan 2024

PONE-D-23-42851Clinical Characteristics and Factors Associated with Long COVID in Zambia, August 2020 to January 2023PLOS ONE

Dear Dr. Malambo,

Thank you for submitting your manuscript to PLOS ONE. After careful consideration, we feel that it has merit but does not fully meet PLOS ONE’s publication criteria as it currently stands. Therefore, we invite you to submit a revised version of the manuscript that addresses the points raised during the review process.

We look forward to receiving your revised manuscript.

Kind regards,

Alanna Gomes da Silva, PhD

Editora Acadêmica

PLOS ONE

Journal Requirements:

“This study has been supported in part by the President's Emergency Plan for AIDS Relief (PEPFAR) through the Centers for Disease Control and Prevention (CDC). Grant number/CoAg ID number: GH002234.

The study sponsor or funder had no role, in the study design; in the collection, analysis, and interpretation of data; in the writing of the report; and in the decision to submit the article for publication. In addition, there is independence of researchers from funders and all authors, external and internal, had full access to all of the data (including statistical reports and tables) in the study and can take responsibility for the integrity of the data and the accuracy of the data analysis.”

“***:**** This study has been supported in part by the President's Emergency Plan for AIDS Relief (PEPFAR) through the Centers for Disease Control and Prevention (CDC). Grant number/CoAg ID number: GH002234”*

“This study has been supported in part by the President's Emergency Plan for AIDS Relief (PEPFAR) through the Centers for Disease Control and Prevention (CDC). Grant number/CoAg ID number: GH002234.”

7. Please include your tables as part of your main manuscript and remove the individual files. Please note that supplementary tables (should remain/ be uploaded) as separate "supporting information" files

Additional Editor Comments (if provided):

The article aims to examine the characteristics of patients presenting at specialized post-acute COVID-19 (PAC-19) clinics in Zambia. We further assessed factors associated with long COVID during the initial visit to a PAC-19 clinic and longitudinally among a cohort of patients with ≥2 PAC-19 clinic visits. This is an extremely relevant topic for public health, given the consequences of COVID after acute symptoms, and there is a need for studies to better investigate long COVID. Health services and professionals need to be prepared for the correct diagnosis and treatment.

The article will require extensive review to be considered for publication. I recommend that the authors put in more effort to improve the entire text, making it clearer, better structured, and including the necessary details and methodological rigor.

SUMMARY: Review all content.

Background: The background section needs to highlight the study's relevance and clearly identify the existing knowledge gap that the research aims to address.

Methods: The methods section should be more comprehensive, providing crucial details such as the study population, duration, key variables, and a thorough description of the analyses performed.

Conclusion: Ensure that the conclusion strictly aligns with the study's objective, summarizing the key findings and their implications in a way that directly responds to the research aim.

INTRODUCTION

The introduction is brief and lacks exploration of the topic, failing to address the knowledge gap. It is crucial to clarify the study's relevance and contribution. Are there limited related studies, and if so, why is this important? What advances does the current study offer compared to existing research? Elaborating on these aspects will provide a stronger foundation for the study's significance.

In general, consider incorporating global epidemiological data on long COVID to set the stage for the study. This can provide readers with a broader understanding of the issue before delving into the specific context of Africa and Zambia. The authors also mention “Zambia’s experience on the COVID-19 pandemic, however, is characterized by high numbers of SARS-CoV-2 infections despite relatively low numbers of confirmed COVID-19 cases”. But they don't explain why cases weren't confirmed, was it due to a lack of testing? The information is vague and is important to complete.

The authors further state: “While dozens of seroprevalence studies in Africa document the extent of spread of SARS-CoV-2...”. However, they only cite two studies, review. Once again, I emphasize that the lack of studies should not be the justification for carrying out the research, as even if there were few, it may have been sufficient. To review.

METHODS

It is imperative to enhance the clarity and completeness of the methods section by adhering to the observational study checklist (STROBE). This checklist should encompass the following details: study design, setting, participants, variables, data sources/measurement, bias, study size, quantitative variables, and statistical methods. The STROBE checklist can be accessed at https://www.strobe-statement.org/checklists/.

Review the methods to align with the STROBE checklist, ensuring all items are presented in the same sequence. Additionally, the statistical analysis section lacks coherence in the sequence of information. The authors mention "regression models" in the plural, but only one model was constructed. This needs clarification.

Furthermore, the location of the study is unclear. Was data collected from all the mentioned hospitals? Please review and rephrase for clarity.

Regarding Table 2, it is unclear if the authors conducted tests other than calculating the 95% confidence interval. Was the Chi-square test with Bonferroni correction performed? Please provide clarification on this matter.

RESULTS

The statement, "...with data entered in the electronic database were included in the analyses," is more appropriate for the methods section rather than the results.

Regarding the authors' statement, "Of these inpatients, 686 (74.6%) were classified as having had severe illness," it is crucial to define what constitutes severe illness for better understanding.

If available, consider including data on the number of vaccine doses received by patients, as this information could be relevant to the study.

It is not clear whether sex, vaccination status, and referral to a specialized service were included in the final model. The authors should specify whether these factors were part of the final model in the results section.

DISCUSSION

The first paragraph in the discussion should serve as a concise summary of the main results, aligning with the study's objective, which will be elaborated on subsequently. Ensure that the discussion follows the sequence of the results.

In the third paragraph concerning the length of stay, there is a need for a more in-depth exploration. Provide a theoretical basis and discuss the findings in the context of existing scientific literature before resorting to comparisons with other studies.

For the fourth paragraph, the mention of bias should be removed. The final paragraph of the discussion is the appropriate space to discuss the study's limitations, including any biases.

Regarding the vaccination aspect, expand the discussion by exploring what other studies reveal about the vaccine. Compare and contrast findings with existing literature. Additionally, delve into the question of how the number of vaccine doses might be associated, emphasizing the importance of including and discussing these results. Refer to relevant literature on this topic.

Ensure that all results are adequately discussed. Topics such as comorbidities, severe illness, and limitations in daily activities are pertinent and should be thoroughly explored. This emphasizes the importance of the initial paragraph where authors present and discuss the main results.

LIMITS: The authors omit a crucial point regarding the local representativeness of the data. It's imperative to acknowledge that the findings may not be extrapolated to other locations or even the entire country, given that the study is based solely on a sample from one institution. Additionally, there is ambiguity regarding the specific hospitals from which the data were collected, requiring clarification for transparency. Please, clarify what are the strengths of this work.

CONCLUSION

The conclusion requires careful review to ensure strict alignment with the study's objectives. Therefore, the main findings of the study must initially be added. Afterwards, other questions can be raised.
---

## [Author Response · Author response to Decision Letter 0]

6 Mar 2024

# Comment Details on how comment has been addressed Manuscript line number

1. Please ensure that your manuscript meets PLOS ONE's style requirements, including those for file naming. The PLOS ONE style templates Manuscript has been updated per PLOS guidelines. This included abstract headings to level 1, sentence case for headings, figure and table citations, author affiliation information, among others Entire document

2. Please provide an amended statement that declares *all* the funding or sources of support (whether external or internal to your organization) received during this study, Provided updated funding statement with all funders and included in the cover letter Cover letter

3. Please remove any funding-related text from the manuscript and let us know how you would like to update your Funding Statement Funding statement received from manuscript and updated stamen provided in cover letter Cover letter

4. Your ethics statement should only appear in the Methods section of your manuscript. If your ethics statement is written in any section besides the Methods, please delete it from any other section. Ethics statement moved to methods section 179-188

5. Please include a separate caption for each figure in your manuscript. Separate captions included 199, 205, & 268

6. Please include captions for your Supporting Information files at the end of your manuscript, and update any in-text citations to match accordingly. Please see our Supporting Information guidelines for more information: http://journals.plos.org/plosone/s/supporting-information. Updated supporting information provided and intext citation updated. 256, 209, 262, 286-287, 373-388;

7. Please include your tables as part of your main manuscript and remove the individual files. Please note that supplementary tables (should remain/ be uploaded) as separate "supporting information" files Tables and figures include in main manuscript and removed individual files. Supporting information retained as separate files Entire document

8. Background: The background section needs to highlight the study's relevance and clearly identify the existing knowledge gap that the research aims to address. Background section updated to highlight the study's relevance and clearly identify the existing knowledge. 49-96

9. The methods section should be more comprehensive, providing crucial details such as the study population, duration, key variables, and a thorough description of the analyses performed. Study design, participants, design, variables, study size, and statistical analysis have now been included in the manuscript 98-178

10. Conclusion: Ensure that the conclusion strictly aligns with the study's objective, summarizing the key findings and their implications in a way that directly responds to the research aim. Conclusion updated to align with objectives 360-370

11. The introduction is brief and lacks exploration of the topic, failing to address the knowledge gap. It is crucial to clarify the study's relevance and contribution. Are there limited related studies, and if so, why is this important? What advances does the current study offer compared to existing research? Elaborating on these aspects will provide a stronger foundation for the study's significance. Addressed as described in part (8) above 49-96

12. The authors further state: “While dozens of seroprevalence studies in Africa document the extent of spread of SARS-CoV-2...”. However, they only cite two studies, review. Once again, I emphasize that the lack of studies should not be the justification for carrying out the research, as even if there were few, it may have been sufficient. To review. Statement reviewed as per comment 91-93

13. It is imperative to enhance the clarity and completeness of the methods section by adhering to the observational study checklist (STROBE). This checklist should encompass the following details: study design, setting, participants, variables, data sources/measurement, bias, study size, quantitative variables, and statistical methods. The STROBE checklist can be accessed at https://www.strobe-statement.org/checklists/.

 Addressed as described in part (8) above 49-96

14. Furthermore, the location of the study is unclear. Was data collected from all the mentioned hospitals? Please review and rephrase for clarity. Study site information provided 99-101

15. Regarding [supplementary] Table 2, it is unclear if the authors conducted tests other than calculating the 95% confidence interval. Was the Chi-square test with Bonferroni correction performed? Please provide clarification on this matter. For the longitudinal analysis which involved 5 PAC-19 clinic data points, we corrected for pairwise comparisons using Bonferroni correction. Notes include in the table updated to highlight the change S1 Table

16. The statement, "...with data entered in the electronic database were included in the analyses," is more appropriate for the methods section rather than the results. Statement moved to method’s section 190

17. Regarding the authors' statement, "Of these inpatients, 686 (74.6%) were classified as having had severe illness," it is crucial to define what constitutes severe illness for better understanding. Severe illness rewritten as “severe COVID” and definition provide accordingly. 133-134

18. If available, consider including data on the number of vaccine doses received by patients, as this information could be relevant to the study. Data on vaccine doses included 229-236

19. It is not clear whether sex, vaccination status, and referral to a specialized service were included in the final model. The authors should specify whether these factors were part of the final model in the results section. Sex included in the model Table 2, Table 3, S3 Table and S4 Table

20 The first paragraph in the discussion should serve as a concise summary of the main results, aligning with the study's objective, which will be elaborated on subsequently. Ensure that the discussion follows the sequence of the results. Paragraph updated and written to be concise 294-300

21. For the fourth paragraph, the mention of bias should be removed. The final paragraph of the discussion is the appropriate space to discuss the study's limitations, including any biases. Bias removed from paragraph 4 and moved to limitation paragraphs 347-361

22. Regarding the vaccination aspect, expand the discussion by exploring what other studies reveal about the vaccine. Compare and contrast findings with existing literature. Additionally, delve into the question of how the number of vaccine doses might be associated, emphasizing the importance of including and discussing these results. Refer to relevant literature on this topic. Discussion on vaccination has been expanded also limitation on vaccination dates restricted classification of vaccination status as full, partial, indeterminate, or unvaccinated. 320-327

23. LIMITS: The authors omit a crucial point regarding the local representativeness of the data. It's imperative to acknowledge that the findings may not be extrapolated to other locations or even the entire country, given that the study is based solely on a sample from one institution. Additionally, there is ambiguity regarding the specific hospitals from which the data were collected, requiring clarification for transparency. Please, clarify what are the strengths of this work. Manuscript updated with limitation on representativeness of the study 347-354

24 The conclusion requires careful review to ensure strict alignment with the study's objectives. Therefore, the main findings of the study must initially be added. Afterwards, other questions can be raised Conclusion updated 363-372

---

## [Decision Letter · Decision Letter 1]

25 Apr 2024

PONE-D-23-42851R1Clinical Characteristics and Factors Associated with Long COVID in Zambia, August 2020 to January 2023: A Mixed Methods DesignPLOS ONE

Dear Dr. Malambo,

Thank you for submitting your manuscript to PLOS ONE. After careful consideration, we feel that it has merit but does not fully meet PLOS ONE’s publication criteria as it currently stands. Therefore, we invite you to submit a revised version of the manuscript that addresses the points raised during the review process.

We look forward to receiving your revised manuscript.

Kind regards,

Alanna Gomes da Silva, PhD

Editora Acadêmica

PLOS ONE

Additional Editor Comments:

Dear Malambo,

After the initial review, other reviewers were invited to review the manuscript and we sent it for corrections.

After careful review by myself and the reviewers, we believe that the manuscript has the potential for publication, however, it will require "Major Revision" throughout the text. I request that you respond to all inquiries in a separate document and that all changes in the text be highlighted in a different color.

Several issues need to be addressed: The manuscript needs to undergo grammatical revision by a qualified professional. Additionally, the manuscript is not formatted according to the citation and reference style required by the journal, which adopts the "Vancouver Style." Therefore, citations should be numerical and superscripted after the period at the end of each paragraph. Please carefully review the "Submission Guidelines" criteria, available at: https://journals.plos.org/plosone/s/submission-guidelines#loc-manuscript-organization, and make all necessary corrections.

Reviewers' comments:

Reviewer's Responses to Questions

**Comments to the Author**

1. If the authors have adequately addressed your comments raised in a previous round of review and you feel that this manuscript is now acceptable for publication, you may indicate that here to bypass the “Comments to the Author” section, enter your conflict of interest statement in the “Confidential to Editor” section, and submit your "Accept" recommendation.

Reviewer #1: (No Response)

Reviewer #2: All comments have been addressed

2. Is the manuscript technically sound, and do the data support the conclusions?

Reviewer #1: Partly

Reviewer #2: (No Response)

3. Has the statistical analysis been performed appropriately and rigorously? 

Reviewer #1: No

Reviewer #2: Yes

4. Have the authors made all data underlying the findings in their manuscript fully available?

Reviewer #1: Yes

Reviewer #2: Yes

5. Is the manuscript presented in an intelligible fashion and written in standard English?

Reviewer #1: Yes

Reviewer #2: Yes

6. Review Comments to the Author

**Reviewer #1**: Title:

The article Clinical Characteristics and Factors Associated with Long COVID in Zambia, August 2020 to January 2023: A Mixed Methods Design, presents a very relevant topic in contemporary times. Thank you for the opportunity to review the article and learn from the authors. I make some suggestions so that the article is ready for publication.

I suggest that the study population be addressed in the title and objectives (adults, children, elderly people)?

Abstract:

In the abstract, in the methods section, it is not clear that it is a mixed methods study. What did the authors consider mixed methods? I suggest not using this “mixed methods” term, as it refers to the use of quantitative and qualitative analyses, considering that two types of quantitative analyzes were adopted in the study (cross-sectional and longitudinal). I suggest removing this term throughout the text and also in the title.

The objective of the summary must be the same as that of the introduction. Review page 12, lines 22 to 25. Review page 14, lines 94 to 96.

Introduction:

In the introduction I suggest that you address what the study contributes and how it advances in relation to the others.

Methods:

On page 15, line 102, Study participants, how old were the study participants? This is important information, as it shows the population studied.

On page 15, on line 119, in Study design, as previously stated, I suggest that the term mixed methods should not be used, considering that both analyzes used were quantitative. I suggest modifying it so that it is a cross-sectional and longitudinal study...

On page 15, in Study design, on lines 119 to 129, the variables are described, I suggest that they be removed to the Study variables and definitions item, with the appropriate explanations of when they were used.

On page 15, on line 130, in the Study variables and definitions item, it must be made clear to the reader what the outcomes were and what the explanatory variables were. How were the variables categorized? I suggest adding.

On page 16, on line 153, as described, it is not very clear how to use Pearson's chi-square test considering the asymmetry of the sample.

On page 16, lines 151 to 156, which states The Pearson Chi-square and Kruskal Wallis tests for proportions and Wilcoxon rank sum test for medians were used to estimate the direction and magnitude of association, I suggest reviewing the use of these tests to estimate direction and magnitude of the association, the association measures of the regression models used allow this, but the use of these tests does not, what allows us to analyze are the presence of differences between strata. However, was a post-test used for the square to locate where the difference is?

On page 16, in lines 157 to 164, it was not clear whether multivariate analysis considered adjustment for all variables, or whether the authors used adjusted analysis for only some variables. I believe it was a multivariate using the p<0.2 criterion for entering variables into the model, but the way it is described is confusing. Need to review and adjust. IN the multivariate analysis in the final model, what was considered statistically significant (p<0.05)? Make that clear.

On page 16, in lines 157 to 177, the association measures extracted by the models used are not included, both for the cross-sectional and longitudinal analyses, this information is only found in the results.

Results:

I suggest that the titles of tables and figures include the population studied, were the patients children, adults, elderly people? ?Include.

In the legend of table 1, some information contained in the legend needs to be included in the methods, such as the significance adopted to analyze the associations using the tests adopted. The question of the denominator and what was calculated.

On page 20, on lines 223 to 221, where are these results presented? Signal.

On page 21, on lines 222 to 228, where are these results presented? Signal.

On page 21, on lines 229 to 236, where are these results presented? Signal.

On page 21, on lines 237 to 243, where are these results presented? Signal.

This information from table 2 must be included in the methods: and Vaccination status and referral to specialist services were not adjusted for at multivariable analysis due to >10% missingness which resulted in 50.5% listwise deletion.

Throughout the result, I suggest reviewing the use of the term probability, since the Odds Ratio were adopted to arrive at the results. Modify to chance.

Discussion

Throughout the discussion I suggest reviewing the use of the term probability, since the Odds Ratio were adopted to arrive at the results. Modify to chance.

Organize the order of the discussion according to the results.

On page 27, lines 347 to 364, the limitations inherent to cross-sectional and longitudinal studies are not mentioned. Furthermore, it was not clear how representative the sample was for Zambia, as well as the possibility of generalizing these data. These limitations and strengths of this study need to be explored in greater detail, providing a counterpoint.

Regarding the vaccination aspect, the explanations of the vaccines used, depending on their brand, are not clear in the discussion. What is the purpose of this information?

Conclusion

On page 28, line 362, the conclusion does not respond to the objective of the study. Need to review.

References

They are adequate and up to date.

**Reviewer #2: **The article aims to examine the characteristics of patients presenting in specialized post-acute COVID-19 (PAC-19) clinics in Zambia. We further assessed factors associated with long COVID at first visit to a PAC-19 clinic and longitudinally among a cohort of patients with ≥2 PAC-19 clinic visits. This is an extremely relevant topic for public health, given the consequences of Covid after acute symptoms and the need for studies to better investigate long Covid, and health services and professionals need to be prepared for the correct diagnosis and treatment.

The article will require extensive review to be considered for publication. I recommend that the authors work a little harder on the entire text so that it is clearer, better structured, with the necessary details and methodological rigor.

SUMMARY: Review all content.

Background: does not address the relevance of the study and the knowledge gap.

Methods: it is superficial, important information such as study population, period, study variables and which analyzes were carried out were not included.

Conclusion: it is not responding to the objective of the study, it must be strictly aligned with the objective.

INTRODUCTION

The introduction is very succinct, the authors did not explore the topic and do not address the knowledge gap, that is, what is the relevance of carrying out the study, is it just because there are few related studies? What advances does the study make in relation to others that have already been published? What impact could this study have?

In general, I suggest bringing global epidemiological data on long Covid, and then addressing the context of Africa and Zambia. The authors also mention “Zambia’s experience on the COVID-19 pandemic, however, is characterized by high numbers of SAR-CoV-2 infections despite relatively low numbers of confirmed COVID-19 cases”. But they don't explain why cases weren't confirmed, was it due to a lack of testing? The information is vague and is important to complete.

The authors further state: “While dozens of seroprevalence studies in Africa document the extent of spread of SARS-CoV-2...”. However, they only cite two studies, review. Once again, I emphasize that the lack of studies should not be the justification for carrying out the research, as even if there were few, it may have been sufficient. To review.

METHODS

It is necessary to make the methods clearer and more complete, hence the importance of following the checklist for observational studies (STROBE), which must contain the following information: Study design; Setting; Participants; Variables; Data sources/measurement; Bias; Study size; Quantitative variables; Statistical methods. Available at: https://www.strobe-statement.org/checklists/

Review the methods and bring all Strobe items in the same sequence. Furthermore, the statistical analysis part does not have a coherent sequence of information. The authors still talk about “regression models” in the plural, but only one model was made. It is not clear.

Another issue that is not clear was the location of the study, was the data collected in all the hospitals mentioned? Review and rewrite.

Did the authors not perform tests for table 2? Was only the 95%CI calculated? Chi-square with Bonferroni correction was not performed?

RESULTS

The following information is not necessary: “...with data entered in the electronic database were included in the analyses”. Because this is methods and not results.

The authors state: “Of these inpatients, 686 (74.6%) were classified as having had severe illness.” What would severe illness be??? It is not clear.

Do the authors have data on how many vaccine doses patients received? If so, I think it's worth adding.

Question: sex, vaccination and referral to a specialized service were not included in the final model?

DISCUSSION

The first paragraph of the discussion should be a summary of the main results in line with the objective of the study, which will be discussed below.

Discuss according to the sequence of results.

In the third paragraph about length of stay, the authors remain superficial and do not delve deeper into the discussion, do not provide a theoretical basis and only compare with some studies. This comparison is not the main one, the entire discussion must be carried out first by bringing scientific/theoretical evidence and, if necessary, comparison.

In the fourth paragraph, the fact of bias must be removed, as the last paragraph of the discussion must include the limits of the study, being the correct space to talk about bias.

As for vaccination, the paragraph should be better explored. What the other studies bring about the vaccine was similar or divergent, to discuss. Furthermore, the question of how many doses can be associated and therefore the importance of including the results and also discussing the doses, what does the literature bring about this?

Were all results discussed? That is why the importance of the first paragraph where the authors set out the main results and will discuss them. There was a lack of discussion about comorbidities; of serious illness; limitation of daily activities, are relevant topics to be explored.

LIMITS: The authors do not mention the fact that the data are locally representative and cannot be extrapolated to other locations or even the country, as it was only a sample from one institution and it is also not entirely clear in which hospitals the data were collected .

What are the strengths of the study?

CONCLUSION

Review the conclusion, as it must strictly respond to the objectives of the study. Therefore, the main findings of the study must initially be added. Afterwards, other questions can be raised.

The article aims to examine the characteristics of patients presenting at specialized post-acute COVID-19 (PAC-19) clinics in Zambia. We further assessed factors associated with long COVID during the initial visit to a PAC-19 clinic and longitudinally among a cohort of patients with ≥2 PAC-19 clinic visits. This is an extremely relevant topic for public health, given the consequences of COVID after acute symptoms, and there is a need for studies to better investigate long COVID. Health services and professionals need to be prepared for the correct diagnosis and treatment.

The article will require extensive review to be considered for publication. I recommend that the authors put in more effort to improve the entire text, making it clearer, better structured, and including the necessary details and methodological rigor.

SUMMARY: Review all content.

Background: The background section needs to highlight the study's relevance and clearly identify the existing knowledge gap that the research aims to address.

Methods: The methods section should be more comprehensive, providing crucial details such as the study population, duration, key variables, and a thorough description of the analyzes performed.

Conclusion: Ensure that the conclusion strictly aligns with the study's objective, summarizing the key findings and their implications in a way that directly responds to the research objective.

INTRODUCTION

The introduction is brief and lacks exploration of the topic, failing to address the knowledge gap. It is crucial to clarify the study's relevance and contribution. Are there limited related studies, and if so, why is this important? What advances does the current study offer compared to existing research? Elaborating on these aspects will provide a stronger foundation for the study's significance.

In general, consider incorporating global epidemiological data on long COVID to set the stage for the study. This can provide readers with a broader understanding of the issue before delving into the specific context of Africa and Zambia. The authors also mention “Zambia’s experience on the COVID-19 pandemic, however, is characterized by high numbers of SARS-CoV-2 infections despite relatively low numbers of confirmed COVID-19 cases”. But they don't explain why cases weren't confirmed, was it due to a lack of testing? The information is vague and is important to complete.

The authors further state: “While dozens of seroprevalence studies in Africa document the extent of spread of SARS-CoV-2...”. However, they only cite two studies, review. Once again, I emphasize that the lack of studies should not be the justification for carrying out the research, as even if there were few, it may have been sufficient. To review.

METHODS

It is imperative to enhance the clarity and completeness of the methods section by adhering to the observational study checklist (STROBE). This checklist should encompass the following details: study design, setting, participants, variables, data sources/measurement, bias, study size, quantitative variables, and statistical methods. The STROBE checklist can be accessed at https://www.strobe-statement.org/checklists/.

Review the methods to align with the STROBE checklist, ensuring all items are presented in the same sequence. Additionally, the statistical analysis section lacks coherence in the sequence of information. The authors mention "regression models" in the plural, but only one model was constructed. This needs clarification.

Furthermore, the location of the study is unclear. Was data collected from all the mentioned hospitals? Please review and rephrase for clarity.

Regarding Table 2, it is unclear if the authors conducted tests other than calculating the 95% confidence interval. Was the Chi-square test with Bonferroni correction performed? Please provide clarification on this matter.

RESULTS

The statement, "...with data entered in the electronic database were included in the analyses," is more appropriate for the methods section rather than the results.

Regarding the authors' statement, "Of these inpatients, 686 (74.6%) were classified as having had severe illness," it is crucial to define what constitutes severe illness for better understanding.

If available, consider including data on the number of vaccine doses received by patients, as this information could be relevant to the study.

It is not clear whether sex, vaccination status, and referral to a specialized service were included in the final model. The authors should specify whether these factors were part of the final model in the results section.

DISCUSSION

The first paragraph in the discussion should serve as a concise summary of the main results, aligning with the study's objective, which will be elaborated on subsequently. Ensure that the discussion follows the sequence of the results.

In the third paragraph concerning the length of stay, there is a need for a more in-depth exploration. Provide a theoretical basis and discuss the findings in the context of existing scientific literature before resorting to comparisons with other studies.

For the fourth paragraph, the mention of bias should be removed. The final paragraph of the discussion is the appropriate space to discuss the study's limitations, including any biases.

Regarding the vaccination aspect, expand the discussion by exploring what other studies reveal about the vaccine. Compare and contrast findings with existing literature. Additionally, delve into the question of how the number of vaccine doses might be associated, highlighting the importance of including and discussing these results. Refer to relevant literature on this topic.

Ensure that all results are adequately discussed. Topics such as comorbidities, severe illness, and limitations in daily activities are pertinent and should be thoroughly explored. This emphasizes the importance of the initial paragraph where authors present and discuss the main results.

LIMITS: The authors omit a crucial point regarding the local representativeness of the data. It's imperative to acknowledge that the findings may not be extrapolated to other locations or even the entire country, given that the study is based solely on a sample from one institution. Additionally, there is ambiguity regarding the specific hospitals from which the data were collected, requiring clarification for transparency. Please clarify what are the strengths of this work.

CONCLUSION

The conclusion requires careful review to ensure strict alignment with the study's objectives. Therefore, the main findings of the study must initially be added. Afterwards, other questions can be raised.

7. PLOS authors have the option to publish the peer review history of their article (what does this mean?). If published, this will include your full peer review and any attached files.

Reviewer #1: **Yes: **Ana Carolina Micheletti Gomide Nogueira de Sá

Reviewer #2: No

---

## [Author Response · Author response to Decision Letter 1]

9 Jun 2024

Table 1: Responses to Reviewer 1

# DESCRIPTION DETAILS ON HOW COMMENT WAS ADDRESSED MANUSCRIPT LINE NUMBER

 GENERAL 

1. The manuscript needs to undergo grammatical revision by a qualified professional. Additionally, the manuscript is not formatted according to the citation and reference style required by the journal, which adopts the "Vancouver Style." Therefore, citations should be numerical and superscripted after the period at the end of each paragraph. The manuscript has been professionally proofread and the referencing style updated to numerical and superscripted “Vancouver style” citation placed after the period at the end of sentence. Throughout the document

2. The article Clinical Characteristics and Factors Associated with Long COVID in Zambia, August 2020 to January 2023: A Mixed Methods Design, presents a very relevant topic in contemporary times. Thank you for the opportunity to review the article and learn from the authors. I make some suggestions so that the article is ready for publication.

I suggest that the study population be addressed in the title and objectives (adults, children, elderly people)? 

Thank you for pointing this out. We agree with this comment and have, therefore, revised the title to now read as follows: “Clinical Characteristics and Factors Associated with Long COVID among Post-acute COVID-19 Clinic Patients in Zambia, August 2020 to January 2023: A Cross-sectional and Longitudinal Study Design” 

Line 1-2

 ABSTRACT 

3. In the abstract, in the methods section, it is not clear that it is a mixed methods study. What did the authors consider mixed methods? I suggest not using this “mixed methods” term, as it refers to the use of quantitative and qualitative analyses, considering that two types of quantitative analyzes were adopted in the study (cross-sectional and longitudinal). I suggest removing this term throughout the text and also in the title.

 The abstract has been re-written to address concerns raised. The term mixed effects has also been removed Line 21-51

 The objective of the summary must be the same as that of the introduction. Review page 12, lines 22 to 25. Review page 14, lines 94 to 96. Thank you pointing this out. This has been addressed. 

 INTRODUCTION 

 In the introduction I suggest that you address what the study contributes and how it advances in relation to the others. Thank you for this suggestion. We have since provided information on what the study contributes and how it advances in relation to other in the last paragraph of the introduction section. Line 91-101

 METHODS 

 On page 15, line 102, Study participants, how old were the study participants? This is important information, as it shows the population studied. We’ve included information on the age of study participants and indicated it as “all ages”. Line 110

 On page 15, on line 119, in Study design, as previously stated, I suggest that the term mixed methods should not be used, considering that both analyzes used were quantitative. I suggest modifying it so that it is a cross-sectional and longitudinal study... Thank you for pointing this out. We agree with this comment and have, therefore, revised the study to a cross-sectional and longitudinal study design. The title has also been revised accordingly. Line 129

 On page 15, in Study design, on lines 119 to 129, the variables are described, I suggest that they be removed to the Study variables and definitions item, with the appropriate explanations of when they were used. We agree with this suggestion. We’ve since moved the paragraph to the study variables section and approximately explained when they were used. Line 134-159

 On page 15, on line 130, in the Study variables and definitions item, it must be made clear to the reader what the outcomes were and what the explanatory variables were. How were the variables categorized? I suggest adding. Thank you for this suggestion. We have more clearly described the outcome and explanatory variables. Categorization for variables has also been included. Line 134-159

 On page 16, on line 153, as described, it is not very clear how to use Pearson's chi-square test considering the asymmetry of the sample. Thank you for pointing this out. We have now more clearly delineated this statement as relating to the cross-sectional analysis. For the longitudinal analysis, measures of associations tests have been removed in supplementary table S3 due to asymmetry of the sample; the property which is inherent for longitudinal studies. Line 170; S3 Table

 On page 16, lines 151 to 156, which states The Pearson Chi-square and Kruskal Wallis tests for proportions and Wilcoxon rank sum test for medians were used to estimate the direction and magnitude of association, I suggest reviewing the use of these tests to estimate direction and magnitude of the association, the association measures of the regression models used allow this, but the use of these tests does not, what allows us to analyze are the presence of differences between strata. However, was a post-test used for the square to locate where the difference is? Thank you for this important comment. We have rephrased this sentence to correctly read as “the Pearson Chi-square and Kruskal Wallis tests for proportions and Wilcoxon rank sum test for medians were used to determine the association of explanatory variables to the outcome”. Statistically significant association were considered at p<0.05. No post-tests were subsequently conducted. Line 172-174

 On page 16, in lines 157 to 164, it was not clear whether multivariate analysis considered adjustment for all variables, or whether the authors used adjusted analysis for only some variables. I believe it was a multivariate using the p<0.2 criterion for entering variables into the model, but the way it is described is confusing. Need to review and adjust. IN the multivariate analysis in the final model, what was considered statistically significant (p<0.05)? Make that clear. We have provided a paragraph under bias to explain how missingness in key variables such as vaccinations status and referral to specialist services had listwise deletion >50% when included at multivariable analysis. These variables were thus not controlled for at multivariable analysis. This is mentioned as a limitation in our study. Line 192-198

 On page 16, in lines 157 to 177, the association measures extracted by the models used are not included, both for the cross-sectional and longitudinal analyses, this information is only found in the results. Thank you for pointing this out. We have rephrased the headings for Table 1 to delineate it more clearly as a descriptive table for the cross-sectional analysis. Similarly, supplementary table S3 is now more clearly labelled longitudinal analysis. Furthermore, the statistical analysis section has be rewritten to provide more clarity. Line 170-183

 RESULTS 

 I suggest that the titles of tables and figures include the population studied, were the patients children, adults, elderly people? ?Include. Thank you for this suggestion. As can be seen in Table 1, the study participants were of all age groups (from <29 years up to 60+ years). The study was not specific to population age group. We have, however, rephrased our titles in tables and figure to make it clear who the study population where. Throughout the document

 In the legend of table 1, some information contained in the legend needs to be included in the methods, such as the significance adopted to analyze the associations using the tests adopted. The question of the denominator and what was calculated. Thank you for this suggestion. All information, including significant level adopted and definition of variables have been included in the methods section. Line 135-159; Line179-181

 On page 20, on lines 223 to 221, where are these results presented? Signal. Thank you for pointing this out. We have since included Fig 3 as the source of this information. Line 248-250

 On page 21, on lines 222 to 228, where are these results presented? Signal. Thank you for pointing this out. We have since included Fig 3 as the source of this information. Line 248-250

 On page 21, on lines 229 to 236, where are these results presented? Signal. Thank you for pointing this out. We have now referenced Table 1 and S3 Table were this information is drawn from. Line 253

 On page 21, on lines 237 to 243, where are these results presented? Signal. Thank for pointing this out. We have since reference Table 2 and further indicated that is in reference to the cross-sectional analysis. Line 261

 This information from table 2 must be included in the methods: and Vaccination status and referral to specialist services were not adjusted for at multivariable analysis due to >10% missingness which resulted in 50.5% listwise deletion. Thank for this suggestion. This information has been included under bias in the methods section Line 193-198

 Throughout the result, I suggest reviewing the use of the term probability, since the Odds Ratio were adopted to arrive at the results. Modify to chance. Thank you for this suggestion. This has been corrected accordingly Line 261-

 DISCUSSION 

 Throughout the discussion I suggest reviewing the use of the term probability, since the Odds Ratio were adopted to arrive at the results. Modify to chance. Thank you for this suggestion. We have adopted the use of the chance throughout the discussion and conclusion section. Line 324-409

 Organize the order of the discussion according to the results. 

 On page 27, lines 347 to 364, the limitations inherent to cross-sectional and longitudinal studies are not mentioned. Furthermore, it was not clear how representative the sample was for Zambia, as well as the possibility of generalizing these data. These limitations and strengths of this study need to be explored in greater detail, providing a counterpoint. Thank you for providing this suggestion. We have since added a statement on the representativeness of the study in the discussion section Line 391-397

 Regarding the vaccination aspect, the explanations of the vaccines used, depending on their brand, are not clear in the discussion. What is the purpose of this information? Per initial review process, we were asked to provide more details on vaccination since our study found a non-significant association. This paragraph serves to provide context of the vaccine type received and the scale up in vaccination which could explain the observed effects.

 CONCLUSSION 

 On page 28, line 362, the conclusion does not respond to the objective of the study. Need to review. The conclusion has been rewritten to align with the study objectives and study findings Line 399-409

Table 2: Responses to Reviewer 2

# DESCRIPTION DETAILS ON HOW COMMENT WAS ADDRESSED MANUSCRIPT LINE NUMBER

 GENERAL 

1. Background: does not address the relevance of the study and the knowledge gap.

 The background has been re-written to provide a knowledge gap and strengthen the manuscript introduction Line 94-104

2. Methods: it is superficial, important information such as study population, period, study variables and which analyzes were carried out were not included. Thank you for pointing this out. We have since more clearly delineated the study outcome and explanatory variables Line 136-161

 Conclusion: it is not responding to the objective of the study; it must be strictly aligned with the objective. Thank you for pointing this out. We have since revised the conclusion Line 399-409

 INTRODUCTION 

 In general, I suggest bringing global epidemiological data on long Covid, and then addressing the context of Africa and Zambia. The authors also mention “Zambia’s experience on the COVID-19 pandemic, however, is characterized by high numbers of SAR-CoV-2 infections despite relatively low numbers of confirmed COVID-19 cases”. But they don't explain why cases weren't confirmed, was it due to a lack of testing? The information is vague and is important to complete.

The authors further state: “While dozens of seroprevalence studies in Africa document the extent of spread of SARS-CoV-2...”. However, they only cite two studies, review. Once again, I emphasize that the lack of studies should not be the justification for carrying out the research, as even if there were few, it may have been sufficient. To review. The background has been re-written to provide a knowledge gap and strengthen the manuscript introduction Line 94-104

 METHODS 

 It is necessary to make the methods clearer and more complete, hence the importance of following the checklist for observational studies (STROBE), which must contain the following information: Study design; Setting; Participants; Variables; Data sources/measurement; Bias; Study size; Quantitative variables; Statistical methods. Available at: https://www.strobe-statement.org/checklists/ Thank you for suggesting the STROBE approach. We have since adopted this approach and revised the manuscript according. Line 106-210

 Review the methods and bring all Strobe items in the same sequence. Furthermore, the statistical analysis part does not have a coherent sequence of information. The authors still talk about “regression models” in the plural, but only one model was made. It is not clear.

Another issue that is not clear was the location of the study, was the data collected in all the hospitals mentioned? Review and rewrite. Thank you for pointing this. On line 126-129, we have provided a statement with regards to how data was routinely abstracted into an electronic databased that was accessed for this data. Data was collected from all the 13 clinics as described in the study setting. Line 107-111; Line 126-129

 Did the authors not perform tests for table 2? Was only the 95%CI calculated? Chi-square with Bonferroni correction was not performed? Thank you for pointing this out. I believe the table being reference is Table 1 since Table is the regression output for the cross-sectional analysis. For the cross-sectional analysis, not pairwise comparisons were made to warrant a Bonferroni correction. For S3 Table however, Bonferroni correction were made S3 table

 RESULTS 

 The following information is not necessary: “...with data entered in the electronic database were included in the analyses”. Because this is methods and not results.

The authors state: “Of these inpatients, 686 (74.6%) were classified as having had severe illness.” What would severe illness be??? It is not clear.

Do the authors have data on how many vaccine doses patients received? If so, I think it's worth adding.

Question: sex, vaccination and referral to a specialized service were not included in the final model? Thank you for pointing this out. We have since removed the quoted text and revised the sentence to read as follows: “Out of a total 1,359 PAC-19 clinic patients in the cross-sectional study (Fig 1), 548 (40.3%) patients with ≥2 PAC-19 clinic visits were included in the longitudinal sub-analysis.” Line 212

Regarding the authors' statement, "Of these inpatients, 686 (74.6%) were classified as having had severe illness," it is crucial to define what constitutes severe illness for better understanding. Thank you for pointing this out. We have rephrased this to correctly read as severe acute COVID-19 Line 42 and 

 If available, consider including data on the number of vaccine doses received by patients, as this information could be relevant to the study. Thank you for this suggestion. A paragraph on number of vaccine doses received by patients have been included. Line 254-262

 It is not clear whether sex, vaccination status, and referral to a specialized service were included in the final model. The authors should specify whether these factors were part of the final model in the results section. Thank you once more for pointing out this. Under bias, we have included statements explaining the reasons the two variables vaccination status and referral to specialized service were not part of the final model. Sex was controlled in the updated model. Line 194-200

 DISCUSSION 

 The first paragraph in the discussion should serve as a concise summary of the main results, aligning with the study's objective, which will be elaborated on subsequently. Ensure that the discussion follows th

---

## [Editor Report · Decision Letter 2]

12 Jun 2024

Clinical Characteristics and Factors Associated with Long COVID in Zambia among Post-acute COVID-19 Clinic Patients in Zambia, August 2020 to January 2023:  A Cross-sectional and Longitudinal Study Design

PONE-D-23-42851R2

Dear Dr. Malambo,

We’re pleased to inform you that your manuscript has been judged scientifically suitable for publication and will be formally accepted for publication once it meets all outstanding technical requirements.

Kind regards,

Alanna Gomes da Silva, PhD

Academic Editor

PLOS ONE

Additional Editor Comments (optional):

None

Reviewers' comments:

None

---

## [Editor Report · Acceptance letter]

23 Jun 2024

PONE-D-23-42851R2 

PLOS ONE

Dear Dr. Malambo, 

I'm pleased to inform you that your manuscript has been deemed suitable for publication in PLOS ONE. Congratulations! Your manuscript is now being handed over to our production team.

Kind regards, 

on behalf of

Dr. Alanna Gomes da Silva 

Academic Editor

PLOS ONE